# Mobile Application Use and Loneliness among Older Adults in the Digital Age: Insights from a Survey in Hong Kong during the COVID-19 Pandemic

**DOI:** 10.3390/ijerph19137656

**Published:** 2022-06-23

**Authors:** Chun Yang, Daniel W. L. Lai, Yi Sun, Chun-Yin Ma, Anson Kai Chun Chau

**Affiliations:** 1Department of Geography, Hong Kong Baptist University, Kowloon, Hong Kong; chunyang@hkbu.edu.hk; 2Faculty of Social Sciences, Hong Kong Baptist University, Kowloon, Hong Kong; daniel_lai@hkbu.edu.hk; 3Department of Building and Real Estate, The Hong Kong Polytechnic University, Kowloon, Hong Kong; yi.sun@polyu.edu.hk; 4Research Institute for Land and Space, The Hong Kong Polytechnic University, Kowloon, Hong Kong; 5Department of Psychology, The Chinese University of Hong Kong, New Territories, Hong Kong; ansonckc123@link.cuhk.edu.hk

**Keywords:** older adults, mobile application use, emotional loneliness, social loneliness, COVID-19 pandemic, Hong Kong

## Abstract

Existing literature on the associations between use of mobile applications (i.e., mobile apps) and loneliness among older adults (OAs) has been mainly conducted before the outbreak of the COVID-19 pandemic. Since mobile apps have been increasingly used by OAs during the pandemic, subsequent effects on social and emotional loneliness need updated investigation. This paper examines the relationship between mobile app use and loneliness among Hong Kong’s OAs during the pandemic. In our research, 364 OAs with current use experience of mobile apps were interviewed through a questionnaire survey conducted during July and August 2021, which assessed the use frequency and duration of 14 mobile app types and levels of emotional and social loneliness. The survey illustrated communication (e.g., WhatsApp) and information apps were the most commonly used. Emotional loneliness was associated with the use of video entertainment (frequency and duration), instant communication (duration), and information apps (duration). Association between video entertainment apps’ use and emotional loneliness was stronger among older and less educated OAs. Our findings highlight the distinctive relationships between different types of apps and loneliness among Hong Kong’s OAs during the pandemic, which warrant further exploration via research into post-pandemic patterns and comparative studies in other regions.

## 1. Introduction

Studies that began soon after the outbreak of the COVID-19 pandemic (since early 2020) have shown an increasing number of mental health problems, particularly among older adults (OAs) [1]. The United Nations asserts that without immediate action, there will be a new pandemic of mental health issues [2]. Among various indicators of wellbeing, loneliness was found widespread among OAs, with 20–34% of OAs in China, Europe, Latin America, and the United States of America (USA) reporting feeling lonely [3]. Victor and Yang (2011) identified a “U” shape of loneliness across ages, with older adults demonstrating the highest level of loneliness, in addition to younger adults [4]. Relevant to depression and mortality, loneliness is an important indicator for OAs’ physical and psychological wellbeing (PWB) [3,5]. In general, loneliness refers to a subjective feeling of a lack of companionship. It happens when there is a mismatch in the quantity and quality of the social relationships between what we have and what we want [6]. Loneliness is conceptualized as an emotive and psychological condition consisting of multifaceted dimensions, such as emotional loneliness and social loneliness defined by Weiss’s typology [7,8], and emotional, social, and existential loneliness defined by Tilburg (2021) [9]. Individuals’ aging and life experiences are essential to their cognitions, behaviors, and levels of tolerance of undesired situations. These personal experiences and characteristics affect whether loneliness intensifies with aging, especially in the COVID-19 context [10].

The rapid development of information and communication technology (ICT) has provided new opportunities to boost everyday life rhythms thanks to the mobile internet and online activities through digital devices. Technological advancements introduce new digital devices and mobile applications (mobile apps) that can serve different purposes in life, such as online shopping (e.g., Amazon), instant communication (WeChat), video entertainment (YouTube), and games (Candy Crush). By utilizing various functions of ICT, such as enabling communication, digital interventions have been undertaken with the aim to increase OAs’ quality of life [11]. For example, some of these are via internet-based physical activity intervention [12], digital reminiscing intervention [13], electronic communication [14], and digital videogames [15]. All of the above demonstrated promising results. Existing research demonstrates a generally positive relationship between OAs’ ICT use and their wellbeing [16,17,18]. However, noticeable gaps remain unsolved. First, study results are mixed across different regions and digital device types. For example, while studies in South Korea and Germany found that the use of a smartphone, tablet, or fitness tracker reduced loneliness [18,19], counter-evidence was reported in another study that suggested a nil relationship between iPad usage and loneliness [20]. Second, previous research mainly concentrated on the communication/socialization uses of ICT [21,22,23,24], with limited attention to other ICT products, particularly mobile apps. Third, loneliness is not a mono-faceted concept. Emotional loneliness refers to a subjective negative feeling associated with the absence of a specific desired social companion, while social loneliness refers to a subjective negative feeling associated with a perceived lack of a wider social network [3,11]. ICT use may lead to different outcomes in terms of social and emotional loneliness.

The perceptions of and attitudes toward ICT use are also influenced by the cultural and social contexts embedded in specific localities. For example, Chinese OAs living in communities put considerable emphasis on their relations with neighbors. Due to the ideology of collectivism (i.e., feelings of belonging to large in-groups that care for them [25]), OAs use social media apps a lot in daily life to maintain interpersonal communication and for knowledge sharing [26]. Different use patterns may affect their wellbeing. Involuntary smartphone use can result in negative wellbeing regardless of OAs’ ages [27]. Apps designed without considering OAs, meanwhile, may be overcomplicated and have complex functions, worsening the relationship between ICT and loneliness.

Against that backdrop, this study will assess the relationship between the use of mobile apps (in terms of frequency and duration) and loneliness for OAs aged 55 and above. The associations of different types of mobile apps with emotional and social loneliness will be assessed. Questionnaire surveys (*n* = 364) were conducted in different communities in Hong Kong between June and August 2021. Hong Kong is a Special Administrative Region (i.e., HKSAR) in China with a 68.1% smartphone ownership rate [28]. Instant communication and social media apps are the first and second most used mobile apps in OAs’ daily lives [29]. Yet, OAs in Hong Kong are free to choose different mobile apps, and we hypothesized that studying those they use would be likely to generate more fruitful insights in terms of the associations between ICT use and loneliness.

The survey in this paper was conducted during the COVID pandemic, with the data collection sites determined according to Hong Kong’s elderly peoples’ geographical distribution (Figure 1) [30]. Due to the social distancing rules, various spaces for OAs’ socialization such as community or elderly centers were closed [31]. OAs’ mental health conditions were reported to be negatively affected. The pandemic pushed OAs to rely more on different mobile apps in their daily lives. The unique context had distinctive implications for the relationship between mobile app use and loneliness for OAs. These trends could set part of the “new normal” even after COVID-19 as OAs become adapted to a more mobile app-dependent lifestyle.

## 2. Theoretical Perspectives and Research Hypothesis

### 2.1. Selective Optimization with Compensation (SOC) Relationships between Use of Different Apps and Loneliness of OAs

Successful aging is “a lifelong process of maximizing gains and minimizing losses through the interplay of three processes: selection, optimization, and compensation” [32]. According to Baltes’ and Baltes’ (1990) model of selective optimization with compensation, older people select some domains and disregard others, engaging in behaviors to maximize the engagement in chosen domains and make up for reduced capabilities to maintain adequate functioning in a selected domain [33,34]. The extent to which OAs acquire an improved quality of life through adapting to the use of mobile apps depends on how they perceive a particular app to compensate for unfavorable lifestyle changes required, due to individual and external contextual factors, such as widowhood, physical constraints, or a change of living environment. SOC sheds light on how OAs view the most critical domains in daily life. Based on that, for example, OAs learn how to use mobile apps to compensate for reduced opportunities to meet with family members and friends, particularly when facing physical disabilities or gathering restrictions (COVID-19).

According to SOC, OAs who face a potential lifestyle change may have greater willingness to use mobile apps. Studies show different functions and purposes of mobile app use may lead to different outcomes. Starting with communication apps, Wetzel et al. (2021) reported a positive relationship between the use of 10 smartphone communication apps (such as Facebook, Instagram, and WhatsApp) and loneliness reduction [24]. By classifying communication apps into instant communication and social networking apps, Simons et al. (2022) found instant communication apps (WhatsApp) were positively related to loneliness reduction, whereas social media app use was not [22]. OAs tend to have relatively fixed Facebook friends [35], suggesting a selection process at play as most friends come from their existing contacts. However, Simons et al (2022) suggest connection-oriented social media was used for reading the feeds only. Fang et al.’s (2017) research on Hong Kong’s OAs found that contact with family members moderated the relationship between OAs’ ICT use and psychological wellbeing (PWB) [33]. Hence, empirical evidence has only agreed that the use of communication apps could reduce loneliness by facilitating communication with OAs’ close social partners, and that instant communication apps are more related to loneliness reduction.

To date, relatively little is known in terms of the relationship between use of other types of mobile apps and loneliness, such as recreational apps. Theoretically, apps such as video entertainment and gaming could help OAs to shift their attention by filling their time and keeping them engaged. Though not focused particularly on mobile apps, entertainment studies generally pointed to the positive effects of entertainment app use on loneliness reduction for OAs. For example, Nimrod (2020) found online leisure reduced OAs’ stress [36]. The’s and Tey’s (2019) study on Chinese OAs found frequent games of cards, mah-jong, TV, and radio entertainment are associated with OAs’ lesser feelings of loneliness [37]. Kahlbargh et al. (2011) found OAs playing on the Wii had lower loneliness and a pattern of greater positive mood compared to the television group [38]. Rubenstein and Shaver (1982) proposed television viewing could be used as a substitute for social contacts and to reduce loneliness, although excessive use of television may inhibit social contact and increase loneliness [39]. As another perspective, Fingerman et al. (2021) found OAs’ television viewing was associated with higher loneliness compared to when not watching television [40]. Further research suggested OAs used entertainment as an active coping strategy to shift their attention from their loneliness [40]. Based on the above review, mobile entertainment apps could minimize OAs’ loneliness. However, the relationship may change if they are overly used, meaning OAs fail to optimize their use or become addicted.

Instrumental use of mobile apps, such as online shopping, could help OAs become less dependent on others to fulfill their daily life needs [41]. They may acquire a sense of autonomy when using such apps. However, this may generate mixed outcomes in loneliness, as when OAs rely less on others, their chances for communication and social mingling are reduced [42]. Previous research has suggested OAs’ daily outings, such as visiting physical stores, could be their only chance to socialize with other people, which is especially important if they live alone [43]. Knowles and Hanson (2018) mentioned OAs were worried that online shopping could take business away from local shops [44], driven by a sense of social responsibility and community embeddedness. They concluded that OAs tried to strike a balance between online proficiency and offline social worlds [44]. The degree of selection and optimization is important when OAs use related apps, as that use may minimize their time for face-to-face interactions. An empirical study in China suggested that OAs felt frustrated and stressed after using mHealth apps. They had insufficient tech self-confidence and were unfamiliar with digitized procedures for making medical appointments [45]. Elsewhere, a further study indicated that involuntary and passive mobile app use could lead to decreased wellbeing among OAs [27].

Theoretically, apps that provide information and daily news to OAs can keep them tuned in with what is happening in their neighborhood and society. In this way, OAs become independent and feel strong connections to society, leading to positive emotions and life satisfaction [46]. Nonetheless, empirical studies reported mixed findings. For example, Nimrod’s (2020) study showed no relationship between information use and stress [36]. Similarly, Deal et al.’s (2018) research found cable news watching had no effect on psychological stress, physiological stress, or the cognitive function of OAs [47]. Instead, Fan and Smith (2021) noted the potential for information overload, which can negatively affect wellbeing [48], with reference to the COVID-19 pandemic. OAs in this case failed to select and filter information for their own needs.

Depending on how successfully an OA can adequately select and optimize the mobile app functions that cater to and compensate for their needs, the relationship between OAs’ use of mobile apps and their loneliness varies. Empirical studies have mainly supported the association of loneliness reduction with communication (particularly instant communication) and recreational app use, while with instrumental and informational apps having an unclear relationship.

### 2.2. Different Association with Emotional and Social Loneliness

Emotional and social loneliness relate differently to mobile app use. OAs “care more about experiencing meaningful social ties and less about expanding their horizons. This motivational shift leads to a greater investment in the quality of important social relationships” (Carstensen et al., 2003, p. 107) [49]. Apps used for communication help maintain existing social ties as OAs care about close social partners more than distant ones. On the one hand, different communication apps provide varying functional capabilities for connecting the user with their social circles, which could build a relationship between social loneliness and use of those apps, such as for youngsters [50]. On the other hand, considering OAs’ preference for maintaining rather than expending social connections, such expanding functions might not be prioritized and comprehensively exploited.

Existing literature seems to suggest that emotional loneliness relates more significantly to OAs’ mobile app use than their social loneliness. Furthermore, studies show that OAs who suffer from emotional loneliness (a sense of emptiness [7]) tend to cope with it actively, and their use of technology is a coping strategy (acting as a pass-time and distracting them from their feelings) [14,51,52,53]. In this way, fleeing a sense of emptiness through games, etc., is more within one’s personal control than resolving social loneliness by finding new people on the internet. Nonetheless, the relationship is affected by OAs’ personal traits and experiences. For example, ICT use had no effect on loneliness for OAs who had recently experienced negative life events, who were not motivated, or who did not have the necessary skills (e.g., communication and computer skills) [14]. Likewise, the effect could be reduced for independent OAs with strong egos and who do not feel lonely. On these occasions, the associations of emotional and social loneliness with app use may not be observed since OAs do not have sufficient motivational reserves.

Empirical results were mixed on this theoretical claim. In the experimental research conducted by Fokkema and Knipscheer (2007) [14], internet use by OAs led to a significant reduction in emotional loneliness and increased self-confidence. Yet, it did not show a significant positive or negative impact on social loneliness. However, other studies have shown that social loneliness is also significantly related to ICT use. Pauly et al. (2019), when focusing on portable ICT, found a significant relationship between the use of social functions (e.g., social media and email) and both emotional and social loneliness [23]. In Sum et al.’s (2008) research on Australian OAs, more frequent use of the internet as a communication tool was associated with a lower level of social loneliness. However, more frequent use of the internet to meet people was associated with a higher level of emotional loneliness [21]. As such, it is important to differentiate between using apps for maintaining and for expanding social networks as these will lead to different loneliness outcomes based on how OAs use the corresponding apps.

### 2.3. Integrated Effects of Age- and Education Level-Related Factors on the Relationship between Mobile App Use and Loneliness

The extent to which older people initiate SOC for an improved quality of life depends on their personal competence and stage of aging. For example, a greater willingness to engage in mobile app use is associated with higher technological literacy, which is influenced by the education level and socioeconomic status of OAs. Higher educational attainment may lead to more ICT use, potentially involving enhanced cognitive function or greater ICT-related knowledge, experience, or interest [54,55,56]. In contrast, OAs with a lower education level could have lower self-efficacy and more significant difficulties (such as facing more serious age-related declines) in using mobile apps [57,58]. They could retrieve fewer benefits from using mobile apps. Thus, the relationship between mobile app use and loneliness could be related to OAs’ education level. Empirical studies were supportive of this claim, though insufficient. As mentioned, ICT use had no effect on loneliness for OAs who were not motivated or did not have the necessary skills [14]. Existing studies also reported the significant moderating role of ICT self-efficacy on the relationship between ICT usage and perceived loneliness of OAs [59,60]. These studies implicate effects of education level-related factors on the relationship between ICT use and loneliness. Such studies were concentrated in cities of Mainland China, where ICT use is regulated by specific institutional and political contexts. There is insufficient empirical evidence on the effect of education level-related factors on the relationship between app use and loneliness. 

Theoretically speaking, age-related factors could affect the relationship between app use and loneliness among OAs when aging is associated with various conditions. Older people could become more dependent on ICT in their daily lives due to age-related declines that limit their physical reserves. This depends on their built-up knowledge regarding ICT. In the other direction, older OAs could be less capable of continuing to use ICT under cognitive and physical declines such as vision loss. The effect of age-related declines on ICT use could be particularly prominent if these apps are not elderly-friendly, e.g., not offering big fonts and multimedia. To the best of our knowledge, little research has investigated the integrated effect of age-related factors on the relationship between mobile app use and loneliness of OAs. Instead, several research efforts used psychological wellbeing (PWB) as the dependent variable, in which findings on the effects of age-related factors were mixed [33,61,62,63]. Different government or research funding can be found across countries and regions [64,65], which may influence use of mobile apps by older OAs with physical declines, affecting age-related factors’ impact on the relationship based on the locality, lending support to the SOC. Studies’ different localities, periods, cohorts, and cultural differences could also have contributed to the mixed findings.

## 3. Research Design and Methods

A questionnaire survey was conducted with 364 participants in Hong Kong, with the inclusion criteria of having experience with mobile apps, locally dwelling, and being aged 55 or above [66]. In this study, we focused on community-dwelling OAs who were relatively independent and ambulatory. Participants were recruited through a hybrid mode (due to social distancing rules at different times) from seven elderly centers and one elderly association affiliated to a local university. Social workers, nurses, and staff members working at the elderly centers and elderly association assisted us in contacting potential participants. The contact lasted until the questionnaire data was inputted and complete data was confirmed by the research team. The elderly centers and association are places where OAs spend spare time in the day, commonly close to their homes. They provide activities in the daytime. In Hong Kong, there are also nursing homes that take care of institutionalized older people.

### 3.1. Variables and Measurements

#### 3.1.1. Loneliness

The six-item De Jong Gierveld Loneliness Scale [67] was used, assessing emotional (three items) and social loneliness (three items). This scale was previously validated in Hong Kong [68]. An example statement for emotional loneliness is “I often feel rejected”. An example statement for social loneliness is “There are enough people I feel close to”. Respondents indicated their extent of agreement with each statement on a five-point rating scale from “strongly disagree” to “strongly agree”. The scale was scored (0–6 for loneliness) following Gierveld and Kamphuls [69].

#### 3.1.2. Mobile Application Use

According to IBM (2016), a mobile app is a software application developed specifically for use on small and wireless computing devices such as smartphones and tablets [70]. We identified 14 types of mobile apps according to a comprehensive literature review (Table 1). Both the mobile app use frequency and duration were measured. Frequency was measured through a rating scale with 0 = non-use; 1 = 1 to 2 days/month; 2 = 1 to 2 days/week; 3 = 3 to 5 days/week; 4 = almost every day. Duration was measured through a rating scale with 0 = non-use, 1 = less than 10 min; 2 = 11–15 min; 3 = 16–30 min; 4 = 31–45 min; 5 = more than 45 min.

To explain underlying mechanisms that account for differences in the use frequency and duration, we asked participants their reasons for using and quitting mobile apps (Figure 2 and Figure 3). We also measured the number of family members/relatives and other contact persons an OA gets in touch with through communication apps. OAs’ attitudes toward social mingling (whether an OA would like to seek attention/share personal thoughts or rather be an observer of their surroundings) were also measured by asking them the frequency of actively posting, with 1 = Never; 2 = less than once a month; 3 = 1–3 times a month; 4 = 1–2 times a week; 5 = 3–6 times a week; 6 = 1–5 times a day; 7 = more than 5 times a day.

#### 3.1.3. Demographic Variables 

Binary variables included their marital status (married or not), housing type (public or private), living alone or not, and whether they were recently employed. Continuous variables included their age, education level (1 = none/preschool, 2 = primary school, 3 = secondary school, 4 = high school, and 5 = tertiary institution) and length of residency in recent housing.

### 3.2. Methods of Data Analysis

Pearson’s correlation and multiple linear regression were used to explore the correlations between mobile app use and loneliness using IBM SPSS 27.0, sourced from HKSAR, China. Controlling for demographic variables, regression analysis was conducted only for significant relationships between mobile app use and loneliness found in correlation analyses. These analyses provided insights into which app type was related to which type of loneliness. Moderation analyses were conducted to examine the integrated effects of age- and education level-related factors (e.g., combined effects of ICT self-efficacy, etc.) on the relationship between OAs’ mobile app use and loneliness, using Hayes’ PROCESS Model 1 [71]. As a default setting, the 16th, 50th, and 84th percentiles were utilized as conditioning values to analyze the change in significance between subgroups of age and education level. Age subgroups were cut off at 62, 70, and 77.6 years old, with OAs aged 77.6 years old or above classified as the old-old. Education level subgroups were cut off at two (primary school), four (high school), and five (tertiary education level). Demographic variables were controlled in all path analyses except when investigated.

## 4. Results

### 4.1. Descriptive Statistics on Mobile App Use, Determinants of Use, and Loneliness of Hong Kong’s OAs

The frequency and duration of use of 14 mobile apps are reported in Table 2. In terms of use frequency, the five most frequently used mobile apps were basic functions (M = 3.74, SD = 0.67), instant communication (M = 3.73, SD = 0.75), information (M = 3.03, SD = 1.48), video entertainment (M = 2.68, SD = 1.57), and pandemic (M = 2.45, SD = 1.77) apps. In terms of use duration, the five mobile apps utilized for the longest duration were video entertainment (M = 3.05, SD = 1.91), instant communication (M = 3.03, SD = 1.52), basic functions (M = 2.86, SD = 1.49), information (M = 2.22, SD = 1.66), and online conference (M = 2.09, SD = 2.07) apps.

For communication purposes, 48% of respondents kept regular contact with more than four family members or relatives. In comparison, 60% of respondents communicated with more than four people who were not their family members or relatives (e.g., friends). Thus, in terms of a quantitative understanding, OAs tended to use mobile apps to contact more individuals who were not their family members or relatives. Among OAs who used social media, most respondents (57%) never posted. Their reasons for mobile app use were investigated (Figure 2): the most crucial reasons were “convenient communication with family members” (84%), followed by “interest in mobile app use” (50%), and “some functions must be accessed with mobile device (45%)”. This indicated that self-motivation (i.e., active use) was the major driver for using apps. In particular, 50% of respondents highlighted that they were interested in mobile app use.

**Figure 2 ijerph-19-07656-f002:**
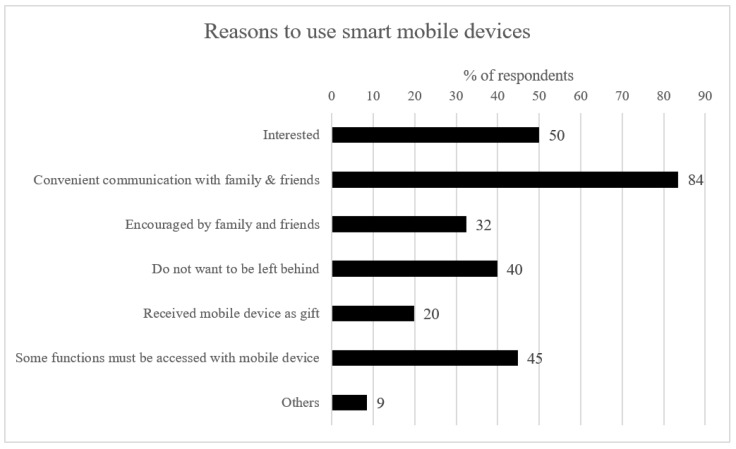
Reasons for respondents to use smart mobile devices. Source: Compiled according to the survey in this study.

In addition, reasons to quit using mobile apps were investigated (Figure 3): the most crucial reasons were “no internet” (22%) and “physical/cognitive decline or not elderly-friendly” (21%). This means attitudinal factors (interesting or not) and physical constraints (capable of using or not) were two reasons for OAs to quit using mobile apps.

**Figure 3 ijerph-19-07656-f003:**
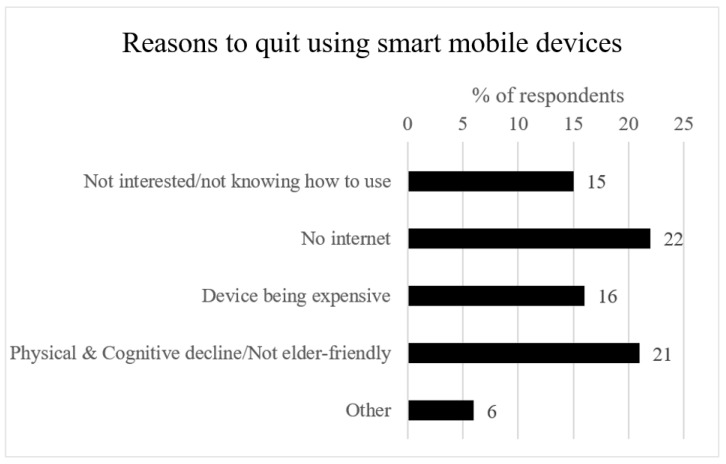
Reasons for respondents to quit using smart mobile devices. Source: Compiled according to the survey in this study.

### 4.2. Relationship between Mobile App Use and loneliness among Hong Kong’s OAs

Correlation and regression analyses were conducted to investigate the relationship between OAs’ mobile app use and loneliness (correlation tables in Appendix A and Table 3). In terms of use frequency, instant communication (r = −0.12, *p* = 0.018), video entertainment (r = −0.16, *p* = 0.003), and information (r = −0.12, *p* = 0.019) apps were significantly and negatively correlated with emotional loneliness. After controlling for demographic variables, only the negative relationship between video entertainment app use and emotional loneliness remained significant (r = −0.08, *p* = 0.008) (Table 3). In other words, more frequent use of a video entertainment mobile app is related to less emotional loneliness. No significant relationship was found between social loneliness and the use frequency of all mobile apps.

In terms of use duration, only instant communication (r = −0.17, *p* = 0.001), mobile payment (r = −0.10, *p* = 0.046), video entertainment (r = −0.14, *p* = 0.007), social media (r = −0.116, *p* = 0.027), and information (r = −0.15, *p* = 0.003) apps were found to be significantly and negatively correlated with emotional loneliness. After controlling for demographic variables, emotional loneliness was only found to be significantly and negative related to instant communication (r = −0.08, *p* = 0.027), video entertainment (r = −0.07, *p* = 0.010), and information (r = −0.07, *p* = 0.035) apps. In other words, longer use of instant communication, video entertainment, and information apps is related to less emotional loneliness. No significant relationship was found between social loneliness and the use duration of all mobile apps.

### 4.3. Integrated Effects of Age- and Education Level-Related Factors on the Relationship between Mobile App Use and Loneliness: Moderation Analysis

Table 4 presents the demographic variables of the survey respondents. When examining the integrated effects of age-related factors, statistically speaking, age negatively moderated the negative relationship between the video entertainment app use frequency and emotional loneliness of OAs on a marginally significant level (β = −0.01, *p* = 0.058) (Figure 4). When applying the 16th, 50th, and 84th percentiles of respondents’ ages (62, 70, and 77.6 years old) (Table 5 and Figure 5a), the negative relationship between video entertainment app use frequency and emotional loneliness was only significant at the 50th (age 70: β = −0.07, *p* = 0.039) and 84th percentiles (age 77.6 (i.e., the old-old): β = −0.12, *p* = 0.001). Integrated effects of age-related factors on the relationship between video entertainment app use frequency and emotional loneliness were only significant for OAs aged 70 or above.

Examining the integrated effects of education level-related factors, statistically speaking, the education level positively moderated the negative relationship between the video entertainment app use frequency/duration and the emotional loneliness of OAs on a statistically significant level (frequency: β = 0.06, *p* = 0.013; duration: β = 0.045, *p* = 0.018). When applying the 16th, 50th, and 84th percentiles of respondents’ education level (primary school, high school, and tertiary education), the negative relationship between video entertainment app use and emotional loneliness was only significant at the 16th percentile (primary education: frequency: β = −0.15, *p* < 0.001; duration: β = −0.12; *p* < 0.001; Figure 5b). The integrated effects of education level-related factors on the relationship between the video entertainment app use frequency/duration and emotional loneliness were only significant for OAs with a primary education level or below.

**Figure 4 ijerph-19-07656-f004:**
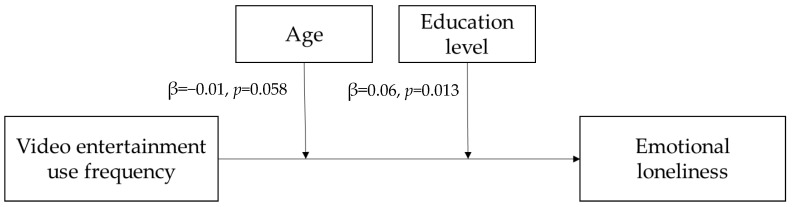
Moderation analysis on the roles of age and education level on the relationship between video entertainment use frequency and emotional loneliness.

**Figure 5 ijerph-19-07656-f005:**
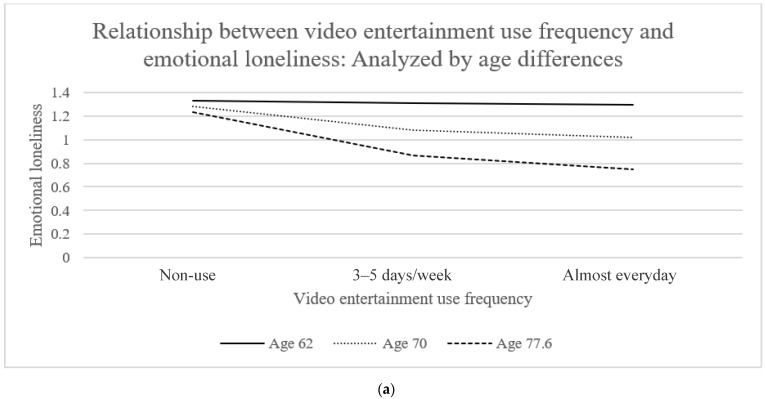
Moderation analysis on the roles of age and education level on the relationship between video entertainment use and emotional loneliness: (**a**) Relationship between video entertainment use frequency and emotional loneliness: Analyzed by age differences. (**b**) Relationship between video entertainment use frequency and emotional loneliness: Analyzed by education level differences.

## 5. Discussion

### 5.1. Mobile App Use Patterns of Hong Kong’s OAs during the COVID-19 Pandemic

Hong Kong OAs have similar use patterns as in other regions [72,73]. The most frequently used apps included basic functions, instant communication, and information apps. Attitudinal factors, including the willingness to try and interest in new things, constitute an important aspect driving the intense use of mobile apps. Hong Kong’s OAs could become more active in learning health-related information by using informational apps in the context of COVID-19.

OAs in Hong Kong seemed unwilling to explore additional functions of mobile apps. One possible reason is that most of Hong Kong’s OAs, particularly those aged 77.6 years old or above (i.e., the old-old) in this study, do not have experience of using smartphones at an earlier stage of life. Apart from physical barriers that obstruct these OAs from using mobile apps, competency factors (e.g., ICT self-efficacy) could also be discouraging them from using particular types of mobile apps. There were other concerns besides their perceived lack of tech-savviness and the difficulties of using mobile apps. For example, some OAs were worried of over-reliance of mobile apps will lead to a loss of activity diversity and miss out interesting experiences in their daily lives, according to Nimrod (2020) [74]. This perception could lead to relatively low use of instrumental apps when the instrumental activities of daily living (IADLs) are digitized.

In particular, there is limited use of instrumental apps—such as using online shopping during times when social distancing measures were in place during the pandemic. In the case of our research, the frequency of using online shopping was ranked as one of the lowest domains. A reason could be that physical shopping trips can offer important social interactions and activities, particularly for those who shop (sometimes daily) largely for the social benefit [44]. Online shopping cannot reach the benefit of social interactions. These concerns are particularly relevant to Hong Kong as many OAs value their connections with neighbors and communities. Our respondents from elderly centers could be considered socially active given the centers offer group activities. They may prefer to interact with others more than average in their daily lives, such as via daily routines and shopping trips. This phenomenon revealed a SOC at play. That is, OAs work hard to strike a positive balance between online proficiency and the cultivation of rich offline social worlds [44].

In terms of use duration, video entertainment and instant communication ranked the top two for Hong Kong’s OAs. With daily activities obstructed, mobile entertainment is a means of killing time. Spending longer before a screen for entertainment was observed elsewhere [75], although it is an unhealthy lifestyle change for OAs. The duration of instant communication was also longer for OAs when using mobile apps as many OAs reduced the times they met their family members in person during COVID-19. Our findings suggest that instant communication apps are used more than other communication app types designed for maintaining or expanding the contact with distanced relations. Examples include email and social media apps. Use patterns across communication apps suggested keeping up communication with meaningful others, such as family and close friends, remains the top reason for using mobile apps. The findings are in line with a previous study suggesting that OAs care most about experiencing meaningful social ties and put considerable emphasis on the quality of social relationships [35]. Moreover, the findings reflect how mechanisms of progress—given the current COVID-19-induced social distancing—have been made to maximize OAs’ chances of keeping contact with close social partners through using communication apps. 

We also found that OAs’ social networks were active throughout times when social distancing measures were in place. In general, OAs kept in touch with fewer family members or relatives compared to other social contacts. As indicated by the more significant relationship of communication app use with emotional loneliness compared to social loneliness, our analysis’ results pointed out that the contact with close family members and relatives had more of an influence on their loneliness. Although connections with other people (including distanced individuals) were of greater quantity, the quality of relationships is the most important thing to OAs. In terms of use patterns for social media, OAs seemed passive, serving as observers rather than actively posting. They preferred to check out others’ activities rather than share their own with distanced contacts. This rather one-directional interaction could have underpinned the lack of significance between social media use and loneliness of OAs.

### 5.2. Relationship between Mobile App Use and Loneliness of Hong Kong’s OAs

Though causal relationships cannot be fully highlighted in our study, our findings on the association between app use and loneliness shed light on future directions for experimental studies aiming to explore causalities. Three findings of this paper have profound implications for policy intervention. First, only 3 out of 14 apps, i.e., instant communication, video entertainment, and information apps, demonstrated significant relationships with the loneliness of OAs. Our findings supplement previous research that provided a general argument that use of ICT is related to OAs’ wellbeing and loneliness, particularly in Hong Kong’s context [18,19]. 

As many OAs today mainly use TV for their entertainment, few researchers have directly analyzed the relationship between mobile entertainment apps and their loneliness. We found that the frequency and duration of using video entertainment apps on their mobile phones was associated with less emotional loneliness. Our findings support previous studies that found associations between ICT use and the wellbeing of OAs [36,37], as well as the effects of ICT use for entertainment on loneliness reduction [39]. To be specific, Nimrod (2020) found increased internet use for leisure was associated significantly with enhanced wellbeing [36]. Teh’s and Tey’s (2019) Chinese study found frequent TV entertainment was associated with OAs feeling less loneliness [37]. Rubenstein and Shaver (1982) asserted that television viewing can to some degree be used to distract from loneliness [39]. Mobile video entertainments’ relationship with OAs’ loneliness was similar to other digital means of entertainment. Although mobile video entertainments are reached with smaller screens that pose more significant challenge to OAs’ visual ability, they received more autonomy on what to watch. Furthermore, video entertainment apps were easy to use in terms of manipulation (search and click) and providing multi-media, meaning OAs could exploit the apps for their own needs. 

However, we did not find a significant relationship between gaming apps and the loneliness of OAs. This contrasts with previous studies that studied either board games (such as cards and mah-jong [37]) or digital games played via other digital devices [38]. In particular, Kahlbaugh et al. (2011) found Wii-playing (a gaming console) was more effective than television watching in reducing OAs’ loneliness [38]. This may be because OAs can hardly play games that require small hand movements on mobile gaming apps due to visual and physical declines. This finding highlights how policymakers have to consider the effects of digital device types (i.e., computer, smartphone, etc.) on the relationship between app use and the wellbeing of OAs, as suggested by McWhorter et al. (2020), following research where an iPad intervention was not effective in reducing OAs’ loneliness [20].

Our research found a significant relationship between instant communication app use (in terms of duration) and emotional loneliness. In other words, our findings only partially supported a previous study where apps for social functioning (such as chat, messaging, email, or social networking, used for finding or contacting community groups, programs, or social events) generally reduced emotional loneliness [23]. We did not find a significant relationship between social media and the loneliness of OAs. This finding seemed to contrast with a previous study that suggested social media, such as Facebook and Instagram, have positive relationships with loneliness reduction [24]. One possible reason might be that most OAs were passive users of social media, who read others’ feeds rather than posting news. 

At the same time, instant communication apps, for the time being, have replaced basic function apps for OAs to use to have deep conversations with meaningful others. Interestingly, instant communication had a less significant relationship with emotional loneliness than video entertainment. As OAs can share information via communication apps, the potential for information overload, leading to more negative wellbeing with specific reference to the COVID-19 pandemic, could be inferred [48]. In the COVID-19 context, this may force OAs into unwanted/undesired social interactions and unwanted forms of sociality. In particular, the quality and content of social engagements are more protective against loneliness in old age compared to quantity, such as the number of friends on the social media platform [4]. Overwhelming conversation poses a challenge for OAs in terms of how to balance receiving selective information and maintaining connections with meaningful social partners.

Long durations of use of information apps, such as news apps, are associated with less emotional loneliness. Referring to information overload, a similar explanation could be given for the less significant relationship of information apps with loneliness of OAs. The study was conducted in the midst of the COVID-19 outbreak, when information outlets were overwhelmed by the pandemic and panic. This posed a significant challenge for OAs in terms of filtering information that could not support their needs. Furthermore, the less significant relationship between information app use and emotional loneliness could be attributed to the multi-purpose nature of video entertainment apps. OAs could reach a wide diversity of videos catering to their varying needs; no matter whether they were seeking information, following sports (e.g., tai chi), etc., entertainment apps provided an “all-in-one function” along with easy functionality. As most OAs aged 77.6 or above (i.e., the old-old) face physical declines and have a lower education level, they may not know how to use more complex and less-elderly-friendly apps. Specifically, information apps may not provide multimedia and may require more work to change the font size. Thus, they could be more dependent on video entertainment apps, as the simplest apps available, to access information, thus reducing their loneliness. Nonetheless, our findings align with theoretical assumptions that information assists OAs’ connection to society, leading to positive emotions and life satisfaction [46]. Our findings stand out as previous studies showed no relationship between information use and OAs’ psychological condition, such as their stress [36] and cognitive function [47]. During the COVID-19 pandemic, Hong Kong OAs could have purposefully selected to receive COVID-19 pandemic-related information, including that on how to confront their loneliness during the pandemic. This information could have helped OAs to stay mentally healthy and safe. As another perspective to understand impacts of COVID-19, OAs in Hong Kong could have understood COVID-19 as a common obstacle, which everyone needed to work together to confront during the pandemic. Instead of information overload, OAs who were selective enough in receiving information could use information apps to build a sense of collectiveness in fighting the virus.

Our study has further implications concerning the relationship between instrumental app use and the loneliness of OAs. Although Tu et al.’s (2021) study of Chinese OAs found involuntary use of mHealth apps was associated with negative wellbeing of OAs [45], we did not find any significant relationship between instrumental app use and loneliness. This was expected considering Hong Kong’s OAs could largely use mobile apps based on personal willingness. A possible meaning may be that instrumental apps sometimes generate mixed outcomes in loneliness, as when OAs rely more on mobile apps, their chances for communication and social mingling are reduced [42].

Drawing on the above discussion, we propose that the frequency and duration of app use may sometimes be associated with different loneliness outcomes. This claim was seldom made in previous studies. Although our research did not measure existential loneliness (i.e., a broader feeling related to the nature of existence and a lack of meaning in life), based on Tilburg’s (2021) typology [9], evidence that ICT facilitates communication and introduces tasks and goals to daily life (e.g., targets in games) implies its role in combatting existential loneliness. As a bidirectional association, an OA with targets and meaning in life may utilize ICT to fulfill their needs, while an OA with high existential loneliness can find the meaning in life via ICT. Nonetheless, as mentioned, previous research has found overuse or heavy use of digital technologies may lead to worsened health and loneliness, such as by forcing OAs into unwanted/undesired social interactions and unwanted forms of sociality [76]. Fan and Smith (2021) proposed that the COVID-19 pandemic information overload and feelings of panic due to the pandemic were related to more negative wellbeing [48]. These studies indicated that a very high frequency and duration of mobile app use may be associated with OAs’ wellbeing and loneliness in a reversed direction. This relates to how adequately OAs can filter excess information from their mobile apps (i.e., selective use based on personal attributes [74]), for example, limiting the amount and type of content they consumed by ignoring and deleting uninteresting messages; quitting annoying mailing lists and groups; blocking advertisements [74]. The COVID-19 pandemic, as an unprecedented occasion, presented a higher requirement for apps’ selective use. 

Our finding that only emotional loneliness significantly related to app use was different from some of the previous research [21,23]. Emotional loneliness is about OAs’ perception of having insufficient meaningful and close social partners, and social loneliness is about the relationship with a broader circle of people; app use seems only to compensate for feelings of alienation from close social partners. This finding echoes another study that noted OAs benefit little from large social networks [19]. In fact, OAs care more about meaningful and close social relations [49]. Moreover, since the pandemic raised the risk of losing a close social partner, OAs might have become more attentive to the situations of their close social partners. The context of COVID-19 could have undermined the role of social loneliness in their relationship with app use.

### 5.3. Integrated Effects of Age- and Education Level-Related Factors on the Relationship between Mobile App Use and Loneliness among Hong Kong’s OAs

The moderation analysis suggested the integrated effects of age- and education level-related factors on the relationship between app use and loneliness. Statistically speaking, significant age and education level moderation effects were only found for the relationship between video entertainment app use and emotional loneliness. A significant relationship between the video entertainment app use frequency and emotional loneliness of OAs was only found for OAs aged 70 or above or with an education level of primary or below. This could imply the OAs aged 70 or above are more inclined to use apps for entertainment purposes. Our finding was consistent with a previous study in Hong Kong [33]. As a potential explanation, video entertainment is very easy to use and does not necessarily require nimble hand movements from the user side (for example, information apps need users to search based on the keywords they can type). As such, age-related declines such as poor eyesight could have less influence on the use experience. The requirement on cognitive abilities will also be lower if OAs prefer to use video entertainment apps for other purposes such as viewing information. 

Significant relationships between video entertainment app use (frequency and duration) and emotional loneliness were found for OAs with an education level of primary school or below. Previous papers suggested that individuals of a lower education level tend to demonstrate technological anxiety [58]. They may prefer more easy-to-use mobile apps such as video entertainment. The findings seem to suggest that OAs with higher and lower education levels develop different ways of using instant communication apps. On a statistical level, we reported an insignificant education level moderation effect on the relationship between information app usage and emotional loneliness. One possible reason might be the attitudinal influence on the degree of usage and its implications for emotional loneliness.

## 6. Conclusions

This paper enriches the developing literature on the relationship between mobile app use and mental health among older people, particularly during the COVID-19 pandemic [77,78], based on a questionnaire survey in Hong Kong. The findings of this paper shed light on how the mobile app use frequency and duration vary largely across Hong Kong’s OAs. We confirm that there is a SOC process to using mobile apps. OAs selected specific mobile app functions and optimized their use through an enhanced frequency and duration. There was widespread use of communication (especially apps providing connections to close social partners) and information apps. Besides this, mobile app use for instant communication, video entertainment, and information was significantly associated with reduced emotional loneliness, but with the causal relationships not yet understood. 

The moderating effects of age and education level were only significant for the relationship between video entertainment app use and emotional loneliness. Specifically, significant relationships between video entertainment app use and emotional loneliness were only found for OAs aged 70 or above and with an education level of primary or lower. The relationships between video entertainment app use and emotional loneliness are related to the age and education level of OAs. This analysis provided insights on the integrated influence of the age and education level (e.g., ICT self-efficacy) on the relationship between app use and loneliness among OAs.

Age-friendly communication and video entertainment apps may be greatly needed in the close future. With OAs worrying about physical and cognitive declines, as shown in our research, governments and organizations can organize more training projects or provide funding to develop apps with bigger fonts. For example, in mainland China, the Ministry of Industry and Information Technology (MIIT) has published guidelines on how websites and mobile app companies can carry out “elderly-friendly modifications”, where compliant sites will receive an official “web accessibility” label [65]. Since some OAs, particularly those aged 77.6 or above (i.e., the old-old), may not be capable of making complex maneuvers, these age-friendly functions should be easy for them to manage. Regardless, however, some OAs may prefer face-to-face interactions or show resistance to ICT as the new mode of face-to-face communication, as reflected in comments from some of the OAs when completing the survey, who still see a sense of human touch as conducive to community social cohesion. Sometimes, OAs may not be able to hear people’s voices or see their faces clearly when communicating through ICT, leaving them feeling they are not capable of adapting to the changing ways of living. Future interventions may consider ways that can familiarize older people with ICT use for daily living. For example, face-to-face tutorials for ICT workshops will be helpful to educate OAs on how to use streaming functions or set devices up for a clear image and sound. In addition, authorities and app developers can consider involving OAs in developing new digital interventions, to better understand their needs and difficulties. OAs are “expert by experience” and are clear on their interests and concerns around using ICT.

Three limitations must be noted to the research presented in this paper. First, a mobile app can serve multiple purposes. For instance, video entertainment apps can be used for pure entertainment and also for information retrieval. Besides this, we used our own rating scale for the use of apps, designed to tap into multi-app use and for easy administration. Our findings warrant further replication with other measures of app usage. Furthermore, the use of a cross-sectional sample limited our inference of the causal relationship between app use and loneliness. As the study was conducted during the interval between the fourth and fifth waves of the COVID-19 pandemic, more comparative studies are needed both locally and internationally to determine whether the patterns observed in the study were time-dependent. The three mobile apps that demonstrated a significant relationship with emotional loneliness were all of high use frequency and duration. Future research should be conducted in a region with a different use pattern. It would then be clear whether the significant relationship was based on use intensity. Furthermore, our study highlighted emotional loneliness as significantly associated with instant communication, video entertainment, and information app use by OAs. There are potential bidirectional relationships between app usage and loneliness. We borrowed insights from Michie et al. (2017) and Western et al. (2021) [79,80], who asserted that digital intervention can be broadly defined as “devices and programs using digital technology to foster or support behavior change”. Various functions of ICT and apps that are conducive to reducing social and emotional loneliness can shed light on new forms and processes of digital intervention targeting mental health promotion. Experimental research is needed to establish the causal relationship between mobile app use and loneliness among OAs in varied locations. 

## Figures and Tables

**Figure 1 ijerph-19-07656-f001:**
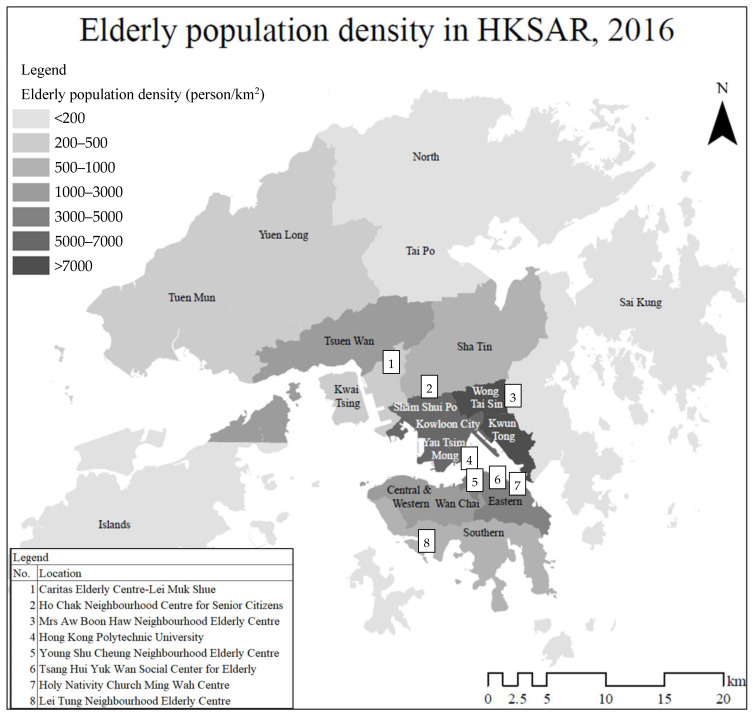
Older adult (OA) population density in Hong Kong (2016) and the locations of survey sites. Source: Adapted from Gong et al. (2016) [30].

**Table 1 ijerph-19-07656-t001:** Major types and examples of mobile apps used among Hong Kong older adults.

Mobile App Types	Examples
1. Basic functions	Phone-call, SMS, Calculator, Radio, Camera
2. Instant communication	WeChat, WhatsApp, Signal, Line
3. Shopping	HKTVmall, Amazon, Taobao
4. Medical service/support	eHealth, HA Go, Personal Emergency Link Service
5. Mobile payment	Alipay, WeChat pay, FPS
6. Video entertainment	YouTube, MyTV, Youku, Iqiyi
7. Social media	Facebook, Instagram, Twitter
8. Information (news/weather)	On.cc, Hong Kong Observatory
9. Financial management	HSBC Mobile Banking, Futubull
10. Outings	HK Taxi, Google Map, KMB 1933
11. Games	Candy Crush, Mahjong
12. Meeting and conferences	Zoom, MS Teams, Tencent Conference
13. Emails	Gmail, Yahoo Mail
14. Pandemic	Leave Home Safe, iAM Smart

Source: Complied based on the survey in this study.

**Table 2 ijerph-19-07656-t002:** Mean and SD on the use of the 14 mobile app types.

Type	Mean Frequency	Mean Duration	Mean Frequency SD	Mean Duration SD
1. Basic functions	3.74	2.86	0.67	1.49
2. Instant communication	3.73	3.03	0.75	1.52
3. Online shopping	0.43	0.71	0.76	1.33
4. Online medical service/support	0.60	0.60	0.88	0.91
5. Mobile payment	0.97	0.65	1.44	1.05
6. Video entertainment	2.68	3.05	1.57	1.91
7. Social media	2.27	1.92	1.78	1.78
8. Information	3.03	2.22	1.48	1.66
9. Financial management	0.76	0.70	1.33	1.31
10. Outings	2.06	1.20	1.57	1.14
11. Gaming	1.39	1.56	1.71	1.94
12. Online conference	0.97	2.09	1.04	2.07
13. Email	1.90	1.49	1.81	1.65
14. Pandemic	2.45	1.24	1.77	1.38
Overall	1.93	1.67	1.12	0.89

**Table 3 ijerph-19-07656-t003:** Multiple linear regression on the relationship between app use and loneliness of OAs.

Application Type	Loneliness Type	r	*p*
Instant communication use duration	Emotional loneliness	−0.080	0.027
Video entertainment use frequency	Emotional loneliness	−0.082	0.008
Video entertainment use duration	Emotional loneliness	−0.065	0.010
Information use duration	Emotional loneliness	−0.067	0.035

**Table 4 ijerph-19-07656-t004:** Demographic variables of the survey respondents.

Characteristics	Values (in % or Mean and SD)
Gender	Male: 25.5% Female: 74.5%
Age	Mean: 70.1 years, SD: 7.51 years
	Range: 55–98 years
Marital status	Single: 10.7%
	Married: 62.6%
	Divorced/separated: 8.2%
	Widowed: 18.4%
Highest level of education	None/preschool: 4.4%
	Primary school: 26.9%
	Secondary school: 17.3%
	High school: 20.6%
	Tertiary institution: 30.8%
Housing type	Public: 49.5%
	Private: 50.5%
Residence length	Mean: 26.0 years, SD: 13.3 years
Living arrangement	Living alone: 23.9% Not living alone: 76.1%
Employment status	Employed: 9.1% Unemployed/retired: 91.9%

Source: Compiled by the authors.

**Table 5 ijerph-19-07656-t005:** Moderation analysis on the roles of age and education level on the relationship between video entertainment use and emotional loneliness.

Application Type	Loneliness Type	Moderator	Age/Education Level	Beta	*p*
Video entertainment use frequency	Emotional loneliness	Age	70	−0.0662	0.0389
Video entertainment use frequency	Emotional loneliness	Age	77.6	−0.1214	0.0011
Video entertainment use frequency	Emotional loneliness	Education level	2 (primary)	−0.1493	0.0003
Video entertainment use duration	Emotional loneliness	Education level	2 (primary)	−0.1187	0.0005

## Data Availability

Ethics restrict us on how we can share the data in this paper, which may contain sensitive information regarding the respondents’ physical and mental health.

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
