# Peer review of "Mobile Application Use and Loneliness among Older Adults in the Digital Age: Insights from a Survey in Hong Kong during the COVID-19 Pandemic"

_ijerph, 2022, doi:10.3390/ijerph19137656_

Round 1

Reviewer 1 Report

Thank you for the opportunity to review this research. I found it to be interesting and clearly written. In accordance with journal guidelines, the paper provides a clear and concise aim for its argument and offers clear context regarding the need for the research. The review of current literature regarding loneliness and mobile application use is adequate. However, I would like the authors to reflect on some issues, namely:

       The authors ought to recognise the “U” shape of loneliness whereby younger and very old people more likely to experience loneliness with it equalling out for mid-life and early retirement. The paper seems to paint too much of picuture of old age= loneliness which overlooks the diversity of older people’s experiences. A good reference to address this point is: Victor CR, Yang K (2012). The prevalence of loneliness among adults: a case study of the United Kingdom. J Psychol;146:85–104.

       Might be worth noting aspects of the ageing experience that also enable coping with loneliness (e.g., life experiences that have promoted resilience). This study about the pandemic specifically but might be useful for this point:
-Birditt, K. S., Turkelson, A., Fingerman, K. L., Polenick, C. A., & Oya, A. (2021). Age differences in stress, life changes, and social ties during the COVID-19 pandemic: Implications for psychological well-being. The Gerontologist, 61(2), 205–216. doi:10.1093/geront/gnaa204

       Authors followed Weiss’s typology of loneliness; I suggest the multidimensional concept of loneliness proposed by Tilburg (2021): besides social and emotional loneliness, the authors could add the concept of existential loneliness (not connecting with others and the world outside, alientation, feelings of isolation, emptiness, and abandonment). Additionally, mortality-related fears were identified to be associated with this type of loneliness, including the fear of disappearing from the earth, the fear of being forgotten, and the fear of dying. This is important when the topic is digital solutions to address loneliness in OAs, particularly considering the pandemic context.

       The methodology section can be improved; for example: who did the recruitment process? Who contacted the participants? How long did the contacts last?

       OAs still prefer face-to-face contacts, as stated in previous studies. There is still some resistance to the use of ICT… could the authors reflect more about these issues and its implications for practice? 

I would also suggest the authors to stress the importance of involving older adults in the design of these digital solutions. They must be considered “experts by experience”. Reflections about the importance of co-design and collaborative/person-centered approaches should be discussed.

Author Response

Many thanks for the constructive comments and suggestions. Please see the attachment for our responses accordingly. 

Reviewer 2 Report

Thank you for the opportunity to review this article. It offers important insights into the use of apps by older people. 

Here are a few comments:

1)

I am not sure if the term "adoption" is really appropriate here.  Adoption means "the decision to start using something". From my point of view, however, this is about use. Frequency and duration also refer to the concept of usage. 

2) 

"Besides, the social, cultural, and political contexts ..." (p.2). It seems as if this could be modelled in the statistical model. However, this is not the case. I recommend deleting the sentence. 

3) 

As a gerontologist, I am generally not very happy if one does not delicately distinguish between genuine age effects (or educational effects) or age-related (or education-related) effects. I would therefore ask you to reconsider the title of chapter 2.3. and some of the wording in the text. 

4) 

In the meantime, the term "likert scale" (p.7) is widely used without question. In my view, however, you are using a rating scale and not the very clear measurement concept of a likert scale - see: Likert, R. (1932). A technique for the measurement of attitudes. Archives of Psychology, 140, 1-55. 

5) 

The presentation of the results should place less emphasis on the correlations (Tab 3 and Tab 4). In my view, these can be included in the appendix. Much more important would be a clear presentation of the regression analysis or moderation analysis. Here, not only tables but also 1-2 diagrams would be useful to visualise the effects. 

6) 

An example statement for emotional loneliness includes "I often feel rejected" An example statement for emotional loneliness includes " (page 6) - I think the second "emotional" loneliness is wrong.   

7) 

I recommend deleting paragraph 3 (lines 215-227) of the conclusion. As stated in the article itself, it is not possible to test causality on the basis of this analysis and many relevant variables could not be controlled. It is highly questionable whether video entertainment really reduces loneliness or whether there is simply an interaction. It is also possible that less lonely people use video entertainment due to social networks, because people want to talk about videos. In this case, the function of use would be completely different.  

Author Response

Many thanks for the constructive comments and suggestions. Please refer to the attachment for the corresponding responses. 

Reviewer 3 Report

I reviewed the article titled:

Mobile Application Adoption and Loneliness among Older Adults in the Digital Age: Insights from a Survey in Hong Kong during the COVID-19 Pandemic   In my opinion the article might be suitable for publishing in International Journal of Environmental Research and Public Health. The authors tackled the very recent problem of loneliness and the influence of mobile apps on the well-being of elderly people. The authors presented very comprehensively the introduction of a subject. The research methodology is clear although quite basic. The results are logical and reliable. Please review the following points prior the publication of this paper:

In the manuscript there is a lack of the authors below the title (which is maybe intentional on this stage of review) and the lines are not numbered, therefore I cannot indicate exactly where the mistakes are. Abstract:
Could you specify how old are older older adults (OAs)? In my opinion this phrase does not sound correctly and it would be better to indicate the age of these people.

Introduction:
What does it mean that the pandemic would change from physical health to a well-being crisis? It would be better to say that without an inmediate action there will be a new pandemic of mental health issues. Please rephrase it.

Could you specify what kind of digital interventions have been undertaken to increase OA's quality of life? The creation of a new application which provides instant communication might enable to be in touch with other people but it does not mean that it was made to reduce loneliness itself. In other words, I would not say that the creation of a new app was a "digital intervention" for loneliness.

First line on the second page: Korea, typo? Or better to say South Korea.

Abbreviation HKSAR should be put in the second page as it is the first mention

Throughout the whole manuscript the authors should change the way of description of "older OAs" or "old-olds" as it is not very precise. In my opinion it would be better to use i.e. seventy-year-olds or eighty-year-olds.

Research Design and Methods:
These centres are the place where elderly people spend their spare time during the day or it is their place where they have been living?

In the table 3 and 4 it would be better to keep only significant results. It would be more readable. Moreover, the font of the table 4 is not correct.

The authors in the discussion section say: "One possible reason might be that online shopping could potentially replace an important social activity for those who shop (sometimes daily) purely for the social benefit." On the other hand if the OAs are living in elderly centres, maybe there's no need to use online shopping apps.

Author Response

Thanks a lot for the very constructive comments and suggestions. Please refer to the attached file for our responses accordingly. 
